



# Editorial: Geoscience communication - Planning to make it publishable

[1]John K. Hillier,[2]Katharine E. Welsh, [3,4]Mathew Stiller-Reeve, [5]Rebecca K. Priestley, [6]Heidi A. Roop, [7]Tiziana Lanza, [8]Sam Illingworth.

[1]Geography and Environment, Loughbrough University, Loughborough, LE11 3TU, UK.
[2]Department of Geography and International Development, University of Chester, Chester, CH1 4BJ, UK.
[3]Konsulent Stiller-Reeve, Valestrandsfossen, Norway.
[4]Center for Climate and Energy Transformation, University of Bergen, Bergen, Norway.
[5]Centre for Science in Society, Te Herenga Waka - Victoria University of Wellington, New Zealand
[6]Department of Soil, Water, and Climate, University of Minnesota, St Paul, Minnesota, USA
[7]Istituto Nazionale di Geofisica e Vulcanologia, Rome, Italy
[8]Department of Learning and Teaching Enhancement, Edinburgh Napier University, Edinburgh EH11 4BN

*Correspondence to*: John K Hillier (j.hillier@lboro.ac.uk)

**Abstract.** If you are a geoscientist doing work to achieve impact outside academia or engaging different audiences with the geosciences, are you planning to make this publishable? If so, then plan. Such investigations into how people (academics, practitioners, other publics) behave can use pragmatic, simple research methodologies accessible to the non-specialist, or be more complex. To employ a medical analogy, first aid is useful and the best option in some scenarios but calling a medic (i.e. a collaborator with experience of geoscience communication or relevant research methods) provides the contextual knowledge to identify a condition and opens up a diverse, more powerful range of treatment options. Here, we expand upon the brief advice in the first editorial of *Geoscience Communication* (Illingworth et al., 2018), illustrating what constitutes robust and publishable work in this context, elucidating its key elements. Our aim is to help geoscience communicators plan a route to publication, and to illustrate how good engagement work that is already being done might be developed into publishable research.

## 1 Introduction

Scientists are increasingly encouraged to have 'impact', effecting real-world changes (e.g. Reed, 2018; Hillier et al., 2019b), which involves communication with non-academic audiences. This communication seeks to involve a range of audiences (e.g. industry leaders, policymakers, students, community groups, indigenous communities, individual citizens) through a variety of activities (e.g. public events, co-writing for social or news media, art installations, classroom visits, workshops). While the interest in scientist-led engagement continues, there are many calls for a closer integration between science communication theory and practice (Salmon and Roop, 2019; e.g. Salmon et al., 2017), and scholars in the field of science communication have spent decades documenting and developing effective methods and practices (e.g. Cheng et al., 2008; Bucchi and Trench, 2008). Similarly, many practitioners of geoscience engagement have lessons to share from their applied experiences.





When *Geoscience Communication* (*GC*) first launched in 2018, the aims of the journal were (Illingworth et al., 2018) to:


1.  provide wider and more formal recognition for existing and future geoscience communication initiatives; and
2.  better formalise the discipline of geoscience communication.

This formalization included a call for increased evaluation and assessment of geoscience communication efforts through the
use of evaluation instruments and social science methods. In its three years, *GC* has published some excellent research articles, making progress on these aims. The current editors of *GC*, though, see the value of exploring the core aspects of what rigorous, evidenced-based geoscience communication research can look like.

As an initial step to achieving the journal's aim, the first editorial in *Geoscience Communication* (Illingworth et al., 2018)
describes what the editorial team wish a paper to look like, in particular highlighting two requirements of good practice:

1.  all research articles should include qualitative and/or quantitative evidence, and not solely anecdotal reporting; and
2.  all research articles should include an explicitly marked section that considers the ethics of the investigation and should also demonstrate how the research has received ethical clearance from their research institute or professional
50       body.

This editorial expands upon these requirements to provide guidance on what constitutes robust and publishable peer-reviewed research in this journal. We use the term 'geoscience communication' to refer to the range of activities included in *GC*; these fall within a spectrum. At one end is activity-led work that might variously be known as education, outreach, communication,
or engagement (e.g. science theatre as a medium for effective dialogue), and at the other end is curiosity-led research (e.g. how videogames tangentially communicate geoscientific concepts) into how people engage with geoscience. The advice in this paper is based on the experience of the current editors, which includes geoscience research, knowledge exchange, science communication and public engagement with science, geoscience education, and the application of social science methods.

Our target audience is two-fold. First, we wish to encourage those who are already doing excellent geoscience communication work but are not publishing it. Second, we would like to support those with less experience who are eager to publish what they have done or will do, but perhaps have not yet considered how to add the necessary rigour into their work. The desire is to convince the former that publication is worthwhile, cost-effective in terms of time, and achievable, and to facilitate the latter in achieving the required quality of study.






This article starts by making a case for publishing papers relating to geoscience communication work (Section 2). We then outline what makes a geoscience communication study publishable as a research article in a peer-reviewed journal (Section 3) and give a step-by-step guide to designing publishable investigations, including exemplar studies and a suggestion of when it might be best to reach out to more experienced colleagues (Sections 4-7). Finally, we cover ethics to demystify this requirement (Section 8), provide an introductory toolkit of research techniques (Section 9), and discuss how to make your article accessible (Section 10), before finishing with a basic framework for turning geoscience communication work into research articles suitable for publications in *GC* (Section 11).

## 2 Why publish work on geoscience communication?

Publishing work on geoscience communication has value, not just as a journal output in addition to those reporting other geoscientific work, not just to comply with a funder's requirements, nor only as a means of communicating with relevant stakeholders (by reading or co-writing an article). Publishing in a peer reviewed journal also has value in building a field of geoscience communication, which is a mechanism to increase the quality and effectiveness of the communication.

In recent years, it has become desirable, if not required, to incorporate a plan for engagement with non-academic audiences (e.g. practitioners, non-specialist citizens, other publics - see Illingworth (2020a)) into the design of scientific projects. Illustratively, there is demand from funding bodies in various countries (e.g. Australia, USA, UK) for a more effective dialogue to share science, leading to changes and benefits outside academia. Specifically, this demand involves the inclusion and rigorous assessment of activities relating to geoscience communication within competitive funding applications.

In the UK, 'impact' is the term used to describe the influence that underlying research has outside academia (Reed, 2018). The UK governmental funding body, UK Research & Innovation (UKRI; https://www.ukri.org, last access: 22 March 2021), defines impact as:

> "An effect on, change or benefit to the economy, society, culture, public policy or services, health, the environment or quality of life, beyond academia".

This ranges from 'awareness raising' (e.g. through co-working with stakeholders) to policy changes (Reed, 2018). In 2020, for most UKRI grants, a separate 'Pathways to Impact' statement describing the approach that will be taken to deliver impact was discontinued, replaced by a requirement for this to be included within the main body of the application, indicative of a continued increase in the importance of impact. Indeed, one recent large funding scheme (the 'Industrial Strategy Challenge Fund' of GBP 4.7 billion) weights impact only slightly less than research excellence, and in another (the 'Global Challenges Research Fund' of GBP 1.5 billion) it is the main objective (UKRI, 2018, 2017). Similarly, in New Zealand, the 'Unlocking



Curious Minds Contestable Fund', which offers up to $2 million of annual funding for STEM engagement projects (Curious Minds, 2019), requires funded projects to report on how they are measuring 'the success' of the project along with an
'assessment of what the project is achieving' (Curious Minds, 2020). The EU, in initiatives such as Horizon 2020, has 'impact' defined similarly to the UK, but with 'expected impacts' clearly defined in its calls for proposals and integrated as a core evaluation (EC, 2018; Reed, 2020). In the USA the National Science Foundation (NSF) includes the potential of the research to achieve societally relevant outcomes within its 'broader impacts' requirement (NSF, 2014).

In many cases, therefore, geoscience communication efforts are already being rigorously designed and evaluated. But, at *Geoscience Communication* we believe that these efforts should be more than a box-ticking exercise to meet funders' requirements. Publishing in a peer-reviewed journal undoubtedly involves significant additional work but, importantly, publication can lead to improved practice (i.e. the work being done better) by drawing upon past work as recorded, and ratified, in previous such journal articles. Now that we have argued that publication is desirable, we consider the characteristics that
make it possible.

### 3 What makes geoscience communication work publishable?

*Geoscience Communication* (*GC*) is a journal that publishes peer-reviewed research. A geoscientist knows what is required to create publishable scientific research within their own core discipline. However, it may not be clear what is involved to do so for a communications activity. So, what makes geoscience work publishable in *GC?* Illingworth et al. (2018) put the advice
very concisely: "All research articles should include qualitative and/or quantitative evidence, and not solely anecdotal reporting". Therefore, research in GC typically consists of the presentation of a research question or hypothesis and the testing of this (i.e. use of the scientific method).

Figure 1 illustrates two ways in which a geoscience communicator can involve research in their practice. In Fig. 1a, the
communication activity is at the fore of the researcher's mind and is subsequently analysed. Here, the research element of the work is overwhelmingly in post-activity evaluation. Alternatively - and preferably - the work is driven by a specific research question (maybe one that is also embedded in previously published work,) and the activity forms part of answering that question (see Fig. 1b). Conceptually, the activity itself could be identical, it is the approach to the project that differs. To link this with something familiar to many geoscientists, consider the approaches to improving a 12-week module for geoscience
undergraduate students. Signoretta et al (2014) revamped a quantitative methods course in order to test the hypothesis that using visualizations (e.g. maps) would improve learner outcomes. Their approach was research-led (Fig. 1b); a particular activity (delivery of a module) needed improving, but driven by funding through the UK government, the aim was to garner widely applicable insights into how this sort of teaching might be improved across the UK. Specifically, the visualization hypothesis arose from the peer-reviewed pedagogical literature, and the activity of delivering the module was part of the



research plan. Alternatively, they might have adopted an activity-led approach (Fig. 1a). If they had made the same changes based upon a personal view in isolation from an academic (i.e. pedagogical) framework, and then decided to evaluate the impact, the research question might have been paraphrased as 'Did it work?' with the research consisting of an evaluation. This is a valid approach, although it comes with a risk that the outcomes are potentially less useful than they might have been (e.g. if a similar piece of work already exists, or if it is difficult to implement the insights elsewhere if not grounded in a theory
that others recognise).

It is fundamental to note that even if the main interest of the author might be in the communication activity itself, what makes it publishable in a peer-reviewed journal such as GC is research that contains a novel insight. When planning publishable work, we encourage integration of research question development and activity planning into a single process, whichever of these is
dominant within a project. To elaborate on what this means in practice, the next sections expand upon the development of a research-led approach.

**4 A spectrum of geoscience communication**

Publishable geoscience communication can be viewed as falling within a spectrum that is based upon the primary motivation of the lead author. At one end is activity-led work that might variously be known as education, outreach, communication, or
engagement, and at the other end is curiosity-led research into how people engage with geoscience. This is illustrated by the banner at the top of Fig. 2; position on this spectrum reflects which parts of the planning process might be foremost in an author's mind.

Of research articles published in *GC* since mid-2018, roughly half are activity-led, commonly framed as evaluations of a
communications activity. Activities include an ephemeral sculpture (Lancaster, 2020), toolkits for science outreach (Locritani et al., 2020), serious games (Skinner, 2020), and ozone monitoring exercises for use in tertiary and higher education (Ramirez-Gonzalez et al., 2020). In addition to evaluations of how much more an audience understands (i.e. a 'deficit model') the GC editorial team would like to see investigations of the dialogue and the communication process itself (e.g. Illingworth, 2017). Often this sort of insight comes through in narrative or other more imaginative and interdisciplinary approaches to evaluation.

The other half of *GC* papers are broadly curiosity-led research investigations into the processes and mechanisms at work in geoscience communication and how humans (academics, practitioners, other publics) engage with and engage in geoscience. This encompasses a wide spectrum of potential topics (see Illingworth et al., 2018). For example, Hillier et al (2019b) seek to understand what motivates academics to collaborate with, and thus communicate with, industry partners; Hut et al (2019) are
curious as to whether geoscientists are better than the wider public at distinguishing real and computer generated landscapes; and Deves et al (2019) probe the biases in media coverage of seismic risks.



Considering these papers in any detail, however, emphasises that our characterization in Fig. 1 is deliberately simplistic. The structure of projects and how a plan to publish geoscience communication work may be built into them is considered further below.

## 5 Planning for publication

The single planning framework in Fig. 2 is applicable wherever on the spectrum of motivation (Fig. 2 banner) authors identify their work to lie. *GC* recognises a variety in authors' perspectives, motivations, resources, and experience, and also that they may be (or have been) more or less cognizant of application and impact (left) or research (right). In practice this means that *GC* accepts papers that focus on one part, while encouraging fully integrated studies; an example of such a study is Archer et al (2021b), which assesses a geoscience communication initiative as an activity in itself but does this by using a robust evaluation set into an appropriate theoretical framework of how such initiatives are designed, so that portable lessons can be learnt and applied more widely and theory advanced.

Existing resources, frameworks and tools can provide detailed guidance on planning your communication activities (Cooke et al., 2017; Illingworth, 2017; Salmon and Roop, 2019), but here we focus on the broad steps involved in designing geoscience communication efforts aligned with leading science communication practices and in a way that can facilitate the publication of these efforts. We pull out and emphasise the research process (right), not to separate it, but rather to provide a familiar point of reference for practicing geoscientists while noting some important additions (e.g. ethics).

As you plan your paper for *GC*, consider the process shown in Fig. 2. At its core is a research process much like that which will be familiar to geoscientists in their scientific work (green), but the framing and purpose which surround and guide the research (blue) need a different sort of consideration.

In terms of framing or defining a geoscience communication activity, particularly at the activity-led end of the spectrum, when you plan your communication activity it is important to be clear about your aim and who your audience will be. Are you trying to encourage behaviour change? Raise awareness of a topic, issue or subject? Influence policy? Inspire more students to pursue careers in Science, Technology, Engineering and Math (STEM)? The answers to these questions should influence how you plan your activity, but also how you will gauge its impact and success. Defining your audience is central for understanding how to shape your activities and messages and to identify if you need to find collaborators who can help in various ways (e.g. in project design, by being appropriate intermediaries) - and in some circumstances this is highly recommended (**Section 7**).



For the research element itself (green box in Fig. 2), as you plan your paper for *GC*, consider the following process. It is much like a research process that will be familiar to geoscientists in their scientific work. Be aware, however, that there are a couple of important points you may be unfamiliar with, particularly if it is not dominantly a curiosity-led investigation (e.g. with a more immediate eye on impact or behaviour change):

1. **Define your research question(s).** For curiosity-led work (e.g. Hut et al., 2019), this is a starting point. But, in all cases we highly recommend that you draft this as you plan your geoscience communication activity. If you are planning to evaluate the impact of your activity you should first clearly define what 'success' looks like, i.e. what are you hoping to achieve? If you have carried out your activity already, then ensure you draft a clear research question before you continue with data collection and analysis.

2. **Identify appropriate methods to collect and analyse the data to answer the questions (Section 7 & 8).** Here you need to test your geoscience communication-related hypothesis, or gauge how your activity has been successful. Illustratively, think about what data you need to evaluate your reach, impact, or chosen mode of communication. Alternatively, what sort of analysis, evaluation, or interrogation will you do to determine the effectiveness, or otherwise, of your project?

3. **Ethical approval (Section 9).** This important element likely differs from the research processes that many geoscientists are used to. If your data-gathering methods involve interviewing or collecting data from human subjects, be sure to obtain ethical approval before you start the data-gathering process (and follow required specified ethical practices throughout the research and writing process).

4. **Collect data.**

5. **Analyse the data.**

6. **Write your paper (Section 10).** Remember that the audience of *GC* spans many fields and disciplines. When writing your paper, please endeavour to write clearly and concisely, avoid jargon (see e.g. Venhuizen et al., 2019) and include critical structural elements you will be familiar with (e.g. introduction, methods, results, discussion, conclusions).

The best geoscience communication efforts will be *informed by* research and will *contribute to* research. In the following section we give some examples of this.

## 6 Three Case Studies

To illuminate aspects of the process of creating a publishable piece of geoscience communication, and the framework in Figure 2, three examples published in *GC* have been selected. Examples 1 and 2 illustrate the spectrum of authors' motivation, while the third exemplifies the potential benefits of reaching out to colleagues across disciplines for support and collaboration.



## 6.1 Example 1

Martin Archer and colleagues carried out an interesting exhibit about sonification at a Science Museum in London (Archer et al., 2021a). Their study finds itself securely in the activity-driven end of the project spectrum in Fig. 2. They planned and carried out a geoscience communication activity, and then evaluated its impact. The aim of the installation was to better communicate the dynamic and active nature of space by converting physical phenomena into sounds and allowing visitors to experience them by listening to them. Ultra-low frequency plasma waves emitted from the sun (the 'solar wind') are analogous

to ordinary sound waves, and the authors presented measurements of these for visitors to hear (using headphones) at their installation.

The authors were all qualified natural/space scientists and were also experienced science communicators. The audience and research aims were important, but also of low enough stakes not to warrant wider interdisciplinary input. Here is an overview

of what they did in relation to the step-by-step process above (see green box on Fig. 2):

1. **Define your research question(s):** The authors' overarching research question was whether their soundscape exhibit had had an impact on the people who attended - did it change their conceptions of space and language they used to describe it? They also had a secondary, technical objective to demonstrate some elements of novelty in the approach

they implemented to evaluate the exhibit's impact.

2. **Identify appropriate methods to collect the data to answer the questions:** Their soundscape exhibit was visited by (mainly) young families who were guided around while listening to the audial experiences of space. The authors chose to use 'graffiti walls' to collect data to answer their research questions. The novelty in this method arises from the use of graffiti walls both before and after visiting the exhibit in order to evaluate any change. The other novelty

in their approach was their use of two complementary statistical methods to analyse the changes they observed on the graffiti walls.

3. **Ethical approval:** The authors followed the Ethical Guidelines of the British Educational Research Association (BERA, 2018) and discussed ethical issues with the institutional funders and the Science Museum before the activity was run. Children only partook in the data collection if they were accompanied, and all data were anonymous.

4. **Collect data:** All the data were collected during the four days the exhibit was open. In total, the graffiti walls before and after the soundscape had 535 and 446 responses respectively.

5. **Analyse the data:** In order to identify any change in attitudes the authors needed to analyse and compare the data from the graffiti walls both before and after the soundscape. They chose two different techniques to do this. They firstly applied quantitative linguistics to analyse how the diversity of words used by the participants changed. They

secondly used thematic analysis to find groups of words connected to broader themes.



6. **Write your paper:** Archer and colleagues wrote up the paper with clear descriptions of all the above steps. It is a good example of how a well-designed science communication activity can be evaluated to show that it had a real impact on the audience that experienced it.

## 6.2 Example 2

An example of a 'curiosity-led' research paper is provided by (Hut et al., 2019). Here the authors of the study were inspired to investigate if geoscientific 'experts' were better at identifying unrealistic geological features in the videogames than 'non-experts'.

The idea for the paper was originally conceived by Hut, Illingworth, and Skinner following discussions of the worldbuilding 265 in the videogame *The Legend of Zelda: Breath of the Wild*. After discussing the approach that they wanted to adopt (a quantitative analysis that ranked participants' confidence in identifying geological features that were either real or from a game) they decided that additional input from a statistical and digital visualisation expert would help in the data collection and analysis phase, and so they approached Albers at the start of the project to help co-design and deliver the study.

As a curiosity-led research paper, the focus in planning was not on an activity or audience represented by the blue box in the planning framework (Fig. 2). Here is an overview of what was done according to the step-by-step research process above (green box on Fig. 2):

1. **Define your research question:** The overarching research question was centred on finding out if people without a
background in the geosciences perceive landscapes from game worlds as more realistic compared to those with a background in the geosciences. In answering this question, the authors also wanted to investigate if wrongfully interpreting game world landscapes as real a risk is when aiming to tangentially communicate geoscientific principles through the use of videogames.

2. **Identify appropriate methods to collect the data to answer the questions:** In the initial scoping exercise for this
study, it was decided that an initial quantitative-based approach would be appropriate to begin to answer the research question. It was also envisioned that this study might then lead to further qualitative research to help further unpick the findings of this study, i.e., that geoscientists are slightly better (with statistical significance) at differentiating real geological features from those in a game world.

3. **Ethical approval:** This study was carried out according to the British Educational Research Association's (BERA)
ethical guidelines for educational research, with all of the data in this study fully anonymised. Furthermore, the survey clearly outlined the purpose of the study, the way in which the data would be used, and provided participants with the opportunity to withdraw from the research at any time.



4. **Collect data:** The data was collected using a survey on Google Forms, through which participants were shown a series of images, some of which were real geophysical features and some of which were from a game world. The participants were asked to mark on an ordinal scale how confident they were in their identification, with the benefit of such ordinal scales being that they can incorporate more nuance than a simple dichotomy. The survey itself was advertised both in person at the European Geoscience Union (EGU) General Assembly 2018 in Vienna and via the Twitter accounts of the authors. While there are limitations to this approach, Côté and Darling (2018) have shown that this is an effective approach for reaching a diverse audience.

5. **Analyse data:** The responses to the survey were analysed using a Student's t-test with Bonferroni correction to account for multiple testing. Furthermore, post hoc analyses showed no significant over-representation of gamers among geoscientists. The specific use of this analysis was discussed very early on in the design of the study, and the survey was designed with this in mind.

6. **Write paper**: From the outset this study had been designed with publication in *GC* in mind, and so the authors were able to be guided throughout each of the preceding stages by the Editorial of Illingworth et al (2018). This helped to ensure that there was a well-designed 'fit', which in turn made preparation for publication more straight forward.

### 6.3 Example 3

An example of a paper that is based on, and benefited from reaching out during project planning and by interdisciplinary collaboration is Hillier *et al* (2019b). The authors were motivated to understand exactly how an individual geoscientist's workload (i.e. specified tasks) and incentive structures (i.e. assessment criteria) may act as a key barrier to university–business collaborations, with a focus on natural hazard risk modelling in the insurance sector.

The work was originally conceived by Hillier with a simple, pragmatic aim of creating a 'user guide' to help initiate and nurture a long-term collaboration between an early- to mid-career environmental scientist and a practitioner in the insurance sector. Hillier, however, realised that this output could be more powerful and broadly applicable if grounded in a body of published theory and practice rather than a mainly anecdotal report of the views of his close contacts in the insurance sector. As primarily a geoscientist, Hillier sought initial advice on what might make the work publishable from the *Geoscience Communications* editorial team, then reached out across specialisms (knowledge exchange experts, social scientists, and insurance practitioners). What emerged is a robust mixed-methods piece of curiosity-led research.

Here is an overview of what they did in relation to the step-by-step research process above (green box on Figure 2):

1. **Define your research question(s):** The study was framed by two broad questions: what motivates academics to do specific work, and reciprocally, what might constrain them? Specifically, this work adds novel insight into why motivations arise and how exactly time constraints manifest themselves in behaviours in the presence of impact





requirements. The constraint focussed upon was the time available in an academic geoscientist's working week as understood through their duties and responsibilities. The motivation focussed upon was the appraisal and promotion structure of universities and the importance of 'impact' (e.g. knowledge exchange or geoscience communication) within this.

2. **Identify appropriate methods to collect the data to answer the questions:** A mixed-methods approach was used, based upon freely available textual data. Job specifications and promotion criteria from UK universities provided data on the tasks required, setting the time constraint, while promotion and therefore its requirements were presumed to be a motivation. To augment this, a workshop interpreting and collecting views on these data was conducted and, further opinions incorporated by co-writing the paper with 22 interested academics and practitioners. So, overall, the

approach draws on ideas of reflexivity and action research.

   3. **Ethical approval:** The study was approved by Loughborough University's departmental ethics coordinator. All data were anonymous.

   4. **Collect data:** Textual data were collected during a desk-based analysis, supplemented by a workshop of 27 participants and, in a novel twist, through comments during co-writing the paper with 22 interested co-authors.

5. **Analyze the data:** In order to identify key aspects of the data three relatively simple qualitative techniques were used: (i) word clouds, (ii) thematic analysis, and (iii) interpretation of participants' comments. No sophisticated methods were used to interpret comments if they were unclear, clarification was simply sought during the writing process (co-authors) or semi-structured interviews (other participants).

   6. **Write your paper:** Hillier and colleagues wrote up the paper, and with a breadth of authors it was written to be

intelligible to all of them – geoscientists, social scientists, and insurance practitioners.

## 7 Reaching out & project planning

Reaching out to professional science communicators or social scientists is a good way to engage in high-quality geoscience communication (Illingworth, 2017). As an example, Priestley et al (2019) analysed the content of reflective blogs and a series of surveys completed by learners engaged in an online course about Antarctic geology and history. The main engagement

activity (the online course) was led by a science historian and a geologist, but co-authors with expertise in geoscience education and psychology were invited to do the thematic analysis and contribute to the publication.

You can assess your need for involving outside expertise on a three-fold basis: your experience, the interdisciplinarity of the project, and the stakes (i.e. risk level associated with a mistake either in the project design or the miscommunication of any

results). This is illustrated in Fig. 3 where we plot the interdisciplinarity of a project against the stakes at play. The placement of the different bands is arbitrary and can change with the experience you might have in interdisciplinarity or with working



with particular topics or issues. Where one places a project on Fig. 3 will depend on one's own values, experience, and skill sets.

In the case of simple survey, if you have never conducted one before then you will likely benefit from at least consulting with someone with survey design and ethics expertise. If you are an experienced geoscience communicator, and the nature of the research question is relatively simple (e.g. 'Did it work?'), you might consider proceeding by yourself or with geoscience colleagues. However, for more interdisciplinary projects, i.e. those with a complex theoretical basis, or where the consequence of misinterpretation is high (e.g. where there is direct feed into policy, or where there are focuses on important ethical or

societal issues), you may need a collaborator with experience in social science methodologies and/or publishing in the field of science communication. Moving up the scales, it is critical you should seek interdisciplinary and even intercultural input if you wish to interact with vulnerable individuals (e.g. children) or groups from substantially different cultural backgrounds to your own, as outlined in the next section.

In our first case study (Archer et al., 2021a), the communication activity had low stakes. On the other hand, the use of audial data and the audience of young families made the project rather interdisciplinary. However, the authors had experience in all these fields, so the project "dot" on Fig. 3 could therefore be placed at the lower end of both axes. For the second case study (Hut el al., 2019), the authors felt they needed input from a statistical and digital visualization specialist. The project also had low stakes but would likely appear higher on the interdisciplinary scale in Fig. 3. The final case study (Hillier et al, 2019) had

much higher stakes since it dealt with issues which were policy relevant. The subject spanned science and industry and the project used a range of research methods. So, the project could be placed quite high on both the stakes and interdisciplinary axes on Fig. 3, clearly indicating a benefit to collaborating with experts from other fields even though the lead author (Hillier) has worked both as an academic and in the insurance sector.

Even if your project is considered to have relatively low stakes or not particularly interdisciplinary, you should still consider collaborating with others outside of your immediate field. Collaborations like these can sometimes be challenging, but they are almost always positive and educational for all involved.

**8 List of possible techniques**

An intention of *GC* is that all research articles should include qualitative and/or quantitative evidence, and not solely anecdotal

reporting (Illingworth et al., 2018). Quantitative evaluation, such as answers on a 1-to-5 (i.e. Likert) scale in a questionnaire, are a readily understood and deployed tool (if there are enough people involved), but qualitative evaluation can also be very powerful. This section is intended as a gateway; an illustration of the range of the toolkit that exists for data collection and analysis, providing links to other literature where such methods have been used in relation to the geosciences.



It is not easy to prescribe what a robust dataset looks like because, like in physical science, this depends on the quality of the data and nature of the research problem; there is a place for both qualitative and quantitative research methods which is largely dependent on the nature of the activity, as well as the theoretical perspectives of the researchers. For example, quantitative evaluations are often suitable for evaluating certain activities as they can reach large numbers of participants quickly and easily. However, if there are too few participants (e.g., n=12), the observations cannot be demonstrated to be statistically

robust. However, in other instances, qualitative research is more appropriate (for example, asking participants to reflect on a longer-term intervention) and a sample of 12 substantive interviews could be an appropriate sample number. Often, a blend of methods yields more reliable results.

## 8.1 Methods for data collection

In order to establish which data collection tools geoscience communicators use in their published research, we have reviewed

those that occur in the existing research articles in *GC.* This exercise demonstrates that pre- and post- surveys to measure change or assess participant perception before and after an intervention, communication, outreach, or educational activity were amongst the most popular methods used to collect data. Researchers used a range of question types to create these surveys, for example Likert scales (e.g. Hut et al., 2019); multiple choice (e.g. Noone et al., 2019) and in some cases, open-ended questions to capture the authenticity, richness, depth of response, honesty and candour of the respondent (e.g. Cohen et al., 2013, p. 225;

Cumiskey et al., 2019). Yet beyond this, innovations such as pre- and post- graffiti walls (e.g. Archer et al., 2021a) were utilised where surveys (for example) were found not to be suitable for the activity.

Perhaps the most familiar data collection tool used by geoscientists is that of field notes. Typically, they are used to record observations as evidence to reflect upon with the purpose of achieving a greater understanding of a phenomenon. Field notes

and observations are also utilised by those within geoscience communication research (Illingworth et al., 2018) as a data collection tool. Collections of case studies and vignettes (e.g. Van Loon et al., 2020) are also used to elicit data from participants in the research.

Other familiar data collection tools such as interviews (e.g. Vicari et al., 2019; Budimir et al., 2020) and focus groups (e.g.

Neumann et al., 2018) are used to elicit rich, qualitative data with interviews being more suitable for instances where individual, and more in-depth responses are required and focus groups typically preferable for discussions and gathering a range of viewpoints.

Authors within *GC* also used secondary or existing data sources of geoscience communication to conduct systematic reviews

(e.g. Loroño-Leturiondo et al., 2019) or else used media reports (e.g. Vicari et al., 2019), social media (e.g. Lacassin et al., 2020), and videogames (e.g. McGowan and Scarlett, 2020) and then went on to analyse data from these sources using new

analytical approaches. Of course, depending on the requirements and nature of the research, sometimes, a mixed-methods approach is the most appropriate (e.g. Hillier et al. 2019). Similarly, depending on the demographic of participants, for example schoolchildren, more appropriate methods such as storytelling (e.g. Davis, 2007; Lanza et al., 2014) or drawings (e.g. Özsoy,
2012) may be used.

## 8.2 Methods for data analysis

Similarly to data collection tools, the analytical techniques used by scholars of geoscience communication, are both quantitative and qualitative in approach. Statistical analyses of questionnaire data (e.g. Stephens et al., 2019; Casado et al., 2020) are often used to quantify changes perhaps pre- and post- event or intervention in order to evaluate and quantify whether
an event, outreach, or communication was effective. Statistical analysis can also be used as a tool to explore the analytics offered by social media channels (e.g. Knudsen and Bolsée, 2019; Skinner, 2020); for example, the number of views of a communication on a YouTube channel could be considered to be a proxy for engagement. Other quantitative approaches could include network analysis (e.g. Narock et al., 2019) from which complex patterns in data can emerge.

Textual analysis, in some format, is often the preferred method of qualitative analysis. Whether through thematic analysis (e.g. Illingworth, 2020b), descriptive coding or the analysis of text within secondary data (e.g. Lacassin et al., 2020), these approaches can offer insight and highlight patterns and themes within the written data. In conjunction, quantitative linguistics, whereby the number of times a theme is alluded to within the text, can be a useful method of pattern identification (e.g. Archer and DeWitt, 2021). Some authors have also used self-reflection of their public engagement initiatives (e.g. Beggan and Marple,
2018) to evaluate an event, outreach, or communication.

It is clear that, in the same way geoscience researchers makes use of a wide range of data collection and analytical techniques, so to do geoscience communication researchers. The nature of the research will largely determine the methods and techniques that are most suitable and appropriate for your research, and should be chosen so as to be congruent with your research
methodology.

## 9 Ethics

The first editorial in *GC* (Illingworth et al., 2018, p.4) highlights ethics as a requirement of good practice, stating:

All research articles should include an explicitly marked section that considers the ethics of the investigation and
should also demonstrate how the research has received ethical clearance from their research institute or professional body.





When collecting data by talking to or eliciting information from 'human subjects', it is important to consider the ethics of the research and seek (sometimes required) ethical approval before starting data collection. Often a streamlined procedure is in place at research institutions, the key role of which is to ensure that participants are not being exposed to unnecessary risks as a result of participating in the research (Guillemin and Gillam, 2004). This is, consequently, a safety net protecting authors without them needing to be an expert in ethics.

In Higher Education Institutions, a board or committee dealing with ethics should also exist. Its name will vary between institutions and countries, but its purpose is the same; to review your research proposal to ensure that you have considered and suitably mitigated for a range of ethical scenarios that could arise as a result of your research. This ethics board may place conditions upon its approval, or reject your proposal if they feel it is too ethically challenging (Healey et al., 2013). If institutional approval is not possible, then the ethical guidelines for a country or governing body should be followed (Illingworth et al., 2018). An example of this is the British Educational Research Association (BERA), which provides ethical guidelines for educational research (see e.g. Flewitt, 2005).

Ethical guidelines in social science research are frequently adopted from the biomedical research community (Tiidenberg, 2020) and typically focus on ensuring dignity, justice, and privacy for the research participants (Eynon et al., 2008; Pittaway et al., 2010) through the processes of "informed consent, confidentiality, and anonymity" (Tiidenberg, 2020: p6) to attempt to mitigate any potential harm to the participant as a result of partaking in the research. Though the suitability of this process for social science has drawn some criticism (e.g. Schrag, 2011; Tiidenberg, 2020) the approach is adopted in many countries across the world.

In detail, researchers are usually required to complete an initial form during the ethics approval process. This may prompt them to consider a range of risk factors and offer mitigation strategies, to ensure data will be held securely and to ensure confidentiality will be guaranteed for personal data (e.g. for participants from the EU GDPR regulations must be complied with). Risk factors could include:

- collecting data from participants under the age of 18;
- psychological or emotional distress as a result of the questions being asked;
- potential for disclosure of current, previous, or proposed antisocial or illegal acts of participants or their associates as a result of the questions being asked;
- potential for discussion of personal/sensitive matters that could be harmful to themselves or others; and
- cultural differences between the researcher and participant that may risk creating misunderstanding or causing offence. For example, it is important that researchers consult carefully with indigenous communities concerning the correct protocols and practices that should be observed during any research that involves them.



Along with the ethics application, researchers are required to submit their data collection tools (e.g. questionnaire or interview questions), 'participation information sheets' (or equivalent) and consent forms for review. Participation information sheets

are required to provide potential participants information about why they have been contacted, what will happen if they take part, whether participation is voluntary, how long the survey/interview (for example) might take, if and how they can withdraw their data, the potential benefits and risks of taking part in the research, how their data will be stored, how confidentiality will be maintained, and what will happen to the data they have provided. Essentially, this is to ensure they can make an informed decision about whether to participate in the research nor not, i.e. that 'informed consent' has been obtained by the researchers.


Typically, researchers are now also required to ensure that the data provided to them by participants will be stored securely i.e. using password protection, encrypted files, and/or locked filing cabinets. New data protection rules, the General Data Protection Regulation (GDPR), were brought in during 2018 to protect the data of residents of the European Union countries; therefore if you are collecting personal data from residents of the EU, you must have a legal basis for doing so. For research,

the legal basis is 'processing in the public interest' and researchers must ensure a privacy notice about how the data will be gathered, stored, and reported is included at the start of the research, typically in the participant information sheet.

Once potential participants have read the participant information sheet, they can then make an informed decision about whether to participate in the research or not. If they agree, participants are required to sign a consent form which asks them to confirm

certain aspects before proceeding. Such a form might ask a participant the following questions:

    (i)       that they have read the participation information sheet and had an opportunity to ask questions about the research;

    (ii)     that they understand participation is voluntary;

    (iii)    that their responses will be anonymous; and

    (iv)    that they are willing for their interview to be recorded (if required by the researcher).


To those unfamiliar with the ethical process, it can, at first, appear arduous. However, it is a necessary process designed to reduce harm to your potential participants and to ensure you, as a researcher, have considered as many possibilities that could arise as possible. Guidance and templates are usually offered by the ethics board and rather than being a barrier or delay to the research, the boards should be viewed as supportive and facilitative to the research if ethically possible. As discussed in Section

7, collaborating with others who are more experienced in these processes is also recommend.

## 10 Widely accessible communication of your research

After data collection and analysis to obtain results, it is time to communicate your research to a wider public of interested parties (e.g. industry, policymakers, researchers from other disciplines). How to best communicate complex findings to the



wider public sphere is a key challenge for scientists (e.g. Illingworth et al., 2018), and this is of particular interest to a journal
of science communication like *GC*. Even to a highly educated and scientifically literate public (e.g. the reinsurance sector),
the onus largely remains on you as the researcher to make your paper accessible. Success in this is highly dependent upon the
language you use.

There is an age-old debate on the use of plain language in scientific journals, with at least some consensus on the utility of
plain language summaries to accompany papers (Bredbenner and Simon, 2019; Hauck, 2019). Even if occasional jargon is the
only way you see to effectively communicate within your field of expertise, you should consider whether it can be eliminated
for a journal such as *GC*, where it is potentially problematic for the target audience. Like other similar journals, the word
'communication' implies interdisciplinary research, including topics such as science engagement and dialogue, science policy,
and education, with GC also including recent fields such as science-art collaborations. The readership of such a journal
potentially includes a wide variety of backgrounds, who are unlikely to know each other's jargon. If the use of jargon is
considered unavoidable, you could explain the terms in the text, but you should note that the presence of jargon (pejoratively
'scientific language') has been shown to interfere with readers' ability to fluently process scientific information, even when
definitions of these terms are provided, which in turn affects their interest in and understanding of the science (Shulman et al.,
2020).


The appropriate use of tables, figures, and video can also assist clear communication. Well-presented tables and figures can
help summarize the salient points of your work, making them accessible to different types of users. This could range from
annotated photographs (Fig. 1b of Lancaster, 2020) to the vast array of geovisualization techniques available including
animations and interactive software tools for data exploration (e.g. Smith et al., 2013). Animation and cartoon summaries can
also be used to good effect (Hillier et al., 2019c, a). *GC* also supports the use of graphical and video abstracts, which can be
used to help reach a wider and more diverse audience.

## 11 Take home messages

Effective geoscience communication is a skill to be learnt, developed, and shared. To be able to improve it as a community,
we need a way to share our experiences of effective and ineffective geoscience communication and one way to do this is
through research and publications. We offer the following basic framework as a guide to creating research publications that
can be published in *GC*:

1.  Develop your approach before acting. If you can name the tools or method(s) you intend to use for data collection
    and analysis then this is a good sign.
2.  Work out what you're trying to achieve.



3. Work out who is your audience is (i.e. who is experiencing or accessing the geoscience).

4. Before doing any research make sure that you have ethical approval.

5. By framing and testing a hypothesis, approach geoscience communication in the same way you would approach other geoscientific research! This is what makes work publishable.

6. Ask for advice and support if you are unsure - whether from colleagues experienced in social science methods, your institutions (e.g. ethics board), or the editors of GC.

7. Use appropriate, jargon-free language, with a combination of tables, graphics, animations, and videos for clear communication.

Good luck! And, if you wish to going further and deeper into the theory and practice of geoscience communication please note that much literature and many frameworks exist (e.g. Cooke et al., 2017; Illingworth, 2017; Salmon and Roop, 2019), which we do not attempt to detail here as this paper is meant as a gateway, and not a complete guide.

**Acknowledgements**

We thank Chris King for his input during developing and writing this paper.

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






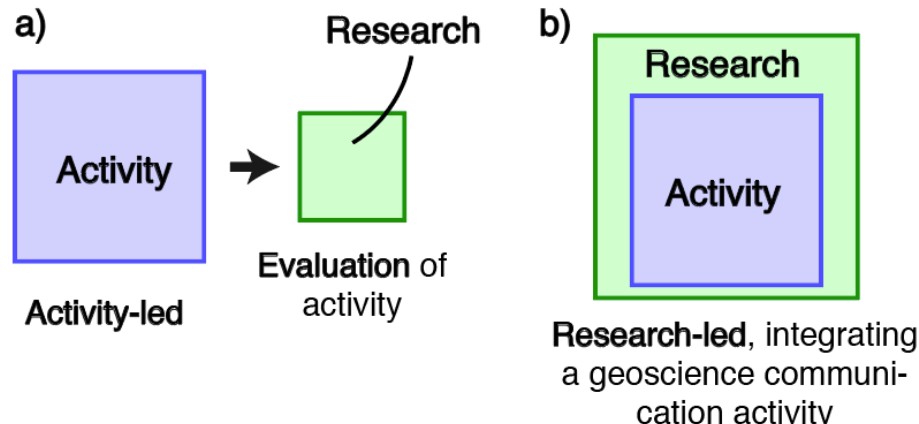

**Figure 1: A conceptual model of two broad approaches to research associated with geoscience communication activities. An integrated approach b) is encouraged, but not obligatory in** *Geoscience Communication*, **and we stress that evaluation of an activity is not the only type of research that is possible.**



**Figure 2: A planning framework for geoscience communication activities, emphasising the presence of the research that makes work publishable (green) within the wider planning framework that makes it useful and impactful. More or less time, weight, or emphasis may be placed on either side, depending upon the authors' resources and motivations, but an integrated approach is encouraged if possible in *Geoscience Communication*. [Section X] annotations in grey indicate relevant sections of this editorial (below).**

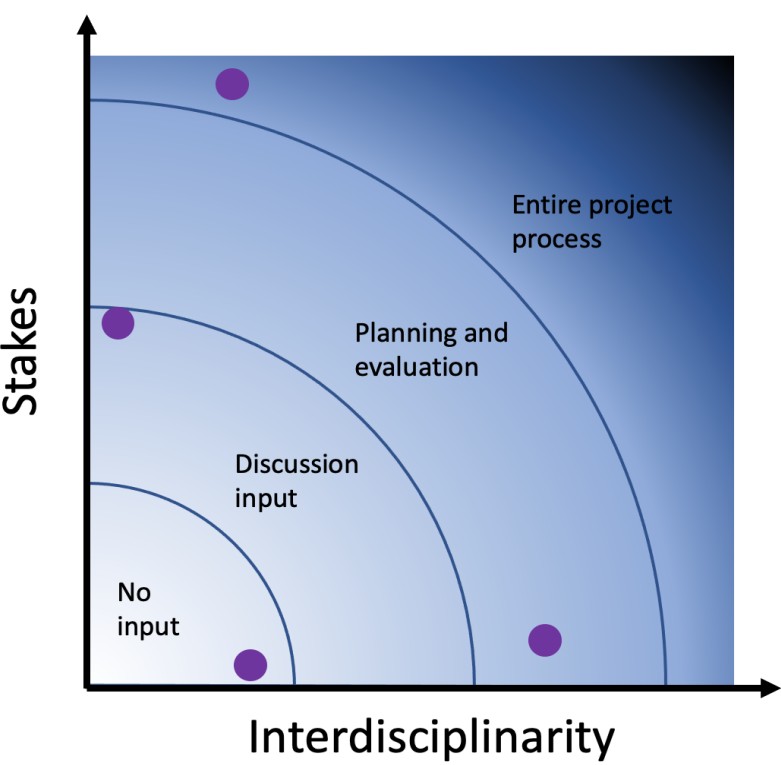

**Figure 3: A typology of project interdisciplinarity (i.e complexity) and stakes (i.e. risk) linked to a zonation of recommendations of when it might be necessary to engage with those outside your geoscience discipline (e.g. social scientists, artists, decision-makers, local communities and so on). Stakes increasing up the y-axis refers to risk of the likelihood and magnitude of a consequence should some error be made increases upwards. On the x-axis, interdisciplinarity increases to the right, and relates to the number of skill-sets required for the project to be a success. The bands in the figure can move according to the researcher expertise in different**
**disciplines or different issues.**