# Peer review of "Editorial: Geoscience communication - Planning to make it publishable"

_Geoscience Communication, 2021_

## Author Comment (AC2)

Replies to comments on **gc-2021-13 "***Editorial: Geoscience communication – Planning to make it publishable***"**

Comments were kindly provided by two reviewers (RC1, RC2) and one member of the geoscience community (CC1). We will use these to improve the editorial.

Please find below our response to the comments. Comments are in grey, and responses in black. Although a fully revised manuscript is not yet prepared, we use 'text changed' (or similar) to indicate some modifications where it was easiest to simply action the comment and provide a document with changes tracked - included at the end of this pdf; in this, comments are used to cross-reference most changes to the number (e.g. RC1.1) assigned to them here.

**RC1 - Martin Archer**

RC1.1 - This editorial provides an incredibly useful guide to the process of undertaking geoscience communication research that might be suitable for publication within the journal GC, and thus the manuscript would make a worthy contribution to the journal itself. The editorial is well structured and outlines key steps, backed up by many published examples, to convince readers that publication of communication activities/research is worthwhile and how one can go about this. I have only minor comments on the manuscript.
> Thank you for your positive and useful review.

General comments

RC1.2 - For activity-driven research, the authors have a tendency to focus only on impact evaluation. Indeed on lines 153-154 the authors state that "the GC editorial team would like to see investigations of the dialogue and the communication process itself", which would constitute a form of process evaluation. This statement comes across as though there are currently no such studies in GC, when this is not the case. Recent examples include: Archer et al. (2021b), Balmer (2021), Skinner (2020). In addition, in my opinion it would also be helpful for the authors to raise the possibility of audience evaluation, i.e. assessing who the audiences of communication activities actually were compared to targets, such as socially disadvantaged demographics (e.g. Archer, 2021) or those that don't typically engage with science (e.g. Archer, 2020). Both of these types of research are often performed in social science and educational research, so would be worth explicitly mentioning somewhere in the article so that a wider range of potential activity-led articles can be understood by readers and potential authors.
> Phraseology modified to avoid the implication that there are currently no such studies in GC, with additional citations also used. The possibility of audience evaluation is now raised.

RC1.3 - I think it would also be helpful for the authors to elaborate on how self-reflection may be used in constructing GC research articles, which is only briefly mentioned on line 434. Many times throughout the manuscript it is stated that qualitative and/or quantitative evidence is required, so reconciling how self-reflection may be included with this statement is required. Self-reflection, grounded in contemporary theory, is a form of Action Research. Mentioning how this may be leveraged in GC would be immensely helpful to potential authors that wish to go down this route.
> We have added the following text to the end of Section 8. It is still brief, but in keeping with the concise nature of this editorial and gives the reader a place to start further reading.

> *For example, you might consider adopting a formal method of reflection (see e.g. Gibbs, 1988; Kolb, 2015) and use this to contextualise your own experiences with that of any feedback that was collated from other researchers and/or participants. Similarly, you might adopt an autoethnographic approach, such as that demonstrated by Reano (2020), in which they engaged in critical reflections of their own practice and lived experiences to reveal how Indigenous research frameworks may enhance the geosciences in higher education.*

Specific comments

RC1.4 - Line 16: "Behave" may be the wrong word, since this implies subsequent actions by participants/audiences. "Respond to these efforts" may be better, since this verb evokes a greater variety of outcomes such as attitudes and thoughts, and also makes clearer the subject of the sentence.
> Text modified.

RC1.5 - Line 26: "may involve" would be more accurate, since there is the possibility of impact that does not include such communication activities or even the active participation of the academic, as evidenced in many REF Impact cases.
> Please clarify.  We cannot see where 'may involve' might fit L26.  Do you mean to modify L25 to "illustrating what robust and publishable work for this journal may involve"?

RC1.6 - Lines 29-32: It would be useful to mention social science and educational research, established fields that have a great amount of overlap with science communication and public/societal engagement, somewhere here.
> We will consider how to best do this. [KEW to consider].

RC1.7 - Line 39: "robust evaluation"?
> Text modified.

RC1.8 - Lines 56 & 278: Perhaps not the right phrase, since "tangential communication" usually refers to going off topic. Maybe "subtly" or "stealthily" communicate would be better?
> "Tangential" is the word selected by the authors of that study (Hut et al, 2019) that this comment is later related to in more detail (i.e. Example 2); "*Video games have a great potential for tangential learning, i.e. learning things about the real world as a tangential benefit while primarily enjoying the experience (Portnow, 2012; Mozelius et al., 2017)*." So we have retained 'tangential'.

RC1.9 - Line 60: "Our target audience for this editorial" in order to clarify that you are not simply referring to the journal's target audience.
> Text modified.

RC1.10 - Line 63: "as well as" instead of the last "and"?
> Text modified.

RC1.11 - Lines 74-75: "other geoscientific work" Please clarify, does this refer to (non- communication) geoscientific research?
> Yes, this refers to the geoscientific analyses, which can subsequently communicated if desired. Text modified.

RC1.12 - Lines 130-135: Personally, I would say this a little harsh on activity-driven research and ignores that qualitatively-drawn conclusions can offer broad insights into why specific aspects may or may not have been well received, which can therefore be applied elsewhere. I would suggest the authors temper this argument slightly.
> L123-135 is autobiographical for the lead author (Hillier), so whilst it might be a bit harsh, it is at least anecdotally true. Had I not been part of the study on the visualization hypothesis, I'd have revised the course based on my training as a lecturer, and then tweaked it further based on *ad hoc* self-reflection and end of module student feedback (i.e. 'Did it work?'). I certainly wouldn't have contributed anything back to the pedagogy of quantitative methods as I wouldn't have reported back to anyone, and I would likely have made some errors identified already in the literature, so the work would have been less useful than it could have been - still useful for the students, but less useful.
> My hope is that this section communicates a clearly as possible two approaches in a familiar situation to many geoscientists in order to encourage them to consider engagement with communication/educational literature more.
> Please let me know if you feel strongly that it's still too harsh. The text now clarifies that the approach I critique is an end-member.

RC1.13 - Line 200: "implementation or impact" to include a broader range of potential research questions?
> Text modified.

RC1.14 - Line 200: The authors should highlight that any success metrics should be, where possible, benchmarked against other available data in published or grey literature and not simply arbitrary.
> Text modified.

RC1.15 - Line 229: "emitted from the sun" is not technically correct since these waves can naturally arise in the solar wind itself as it travels towards Earth or as the wind interacts with Earth's magnetosphere. "due to the 'solar wind'" would be fine.
> Apologies. Text modified.

RC1.16 - Line 233: This sentence is a slight mischaracterisation of the authors. While the statement is true of the first author, the co-authors have different scientific backgrounds (e.g. medical science) but are principally public engagement professionals/practitioners.
> Apologies for inadvertently mischaracterising your co-authors. Text modified.

RC1.17 - Line 234: What the authors mean by "stakes" is not defined until much later, so perhaps should not be referred to at this point in the manuscript.
> Modified to avoid the term 'stakes' here, and added a reference Fig. 3

RC1.18 - Line 270: "or audience" I would suggest this is removed, since there were clear audiences in mind during the planning (geoscientists vs. non-geoscientists).
> We agree with the reviewer that there were clear audiences in mind, but for this editorial we wish to remain very explicit, and retain the words.

RC1.19 - Line 342: "Science communication researchers"? Not all professional practitioners are trained in evaluation/research methods and/or underlying theory.
> Good point. Text modified.

RC1.20 - Lines 349-349: "interdisciplinarity of the project and stakeholders"? In some of the examples presented, the authors already had interdisciplinary expertise that they could leverage in order to enable publication.
> Good point. Text modified to clarify.

RC1.21 - Lines 375-378: Perhaps the authors could comment that the act of collaborating with different disciplines might make authors to GC more skilled in new areas and thus able to continue publishing their communication activities/studies with less assistance in the future?
> Happy to add this comment. Added.

RC1.22 - Line 389: "might not" instead of "cannot" as this will also depend on the effect size.
> Indeed. Changed.

RC1.23 - Lines 418-420: These appear to be primary sources of data, so should they not go on Line 400 along with the mention of graffiti walls (which included drawings as well as words)?
> These are primary sources of data, and would be better mentioned before this last paragraph. They have been moved to the paragraph above - L400 was specifically focussed on pre- post- methods to evaluate an activity.

RC1.24 - Line 418: Raising demographics here highlights the need to discuss demographic data, either as a primary or secondary data source (e.g. Archer, 2020, 2021).
> Demographic data are certainly useful, but we prefer to avoid diving too deep into any particular type of data, and are not claiming to be listing types exhaustively, rather to be illustrating. So, we have not added a specific discussion on demographic data.

RC1.25 - Line 424: "the size and significance of any potential changes" in order to highlight that there may be no real changes from before to after as a result of robust statistical analysis?
> Text modified.

RC1.26 - Line 426: Perhaps add comments and likes alongside views for YouTube videos?
> Text modified.

RC1.27 - Line 432: Quantitative linguistics concerns empirical properties and laws of languages, whereas what the authors refer to here is quantifying qualitative data.
> Text modified.

RC1.28 - Figure 1: Panel a is somewhat misleading, since the collection of evaluative data requires prior-planning and thus the research element cannot be wholly unconnected from the activity. I would suggest the authors modify to include some slight overlap to the activity and research.
> For simplicity, we wish to illustrate end-member approaches to geoscience communication. So, whilst in practice we agree with the reviewer that research cannot be entirely decoupled from the activity, we have not included an overlap.  However, we have clarified that these are end-member viewpoints in the figure caption and main text.

RC1.29 - Figure 2: The numbering does not start at the top left, which may be confusing for readers. I can see that this has been chosen to align with the arrows, however, if I understand correctly, the process is not a cycle thus such cyclical arrows are not warranted. I would suggest the authors use a simpler depiction either using typical flow chart style or even just a vertically numbered list.
> Thank you for this comment. We have amended the figure, striving for a simpler depiction. Words have been reduced, and greyscale adopted (as experimentation with colour schemes showed this was clearer as well as being more accessible e.g. to colour blind readers), and attempting to better display the nuance in the relationship between the boxes (e.g. simpler arrows, bold outline to more strongly connect step (v) and the research box).
> How best to depict the processes associated with the planning of a communication activity and the research inter-twined with it was a subject of intense and protracted debate amongst the *GC* editors co-authoring this paper. The elements colleagues wished to emphasise differed according to their background, and the conceptual simplification presented is synthesis and compromise. A few points relevant to the figure's design are outlined below.

- A numbered list with 'Activity planning' as a pre-cursor step to the 'Research' steps was unacceptable to about half of co-authors.
- A numbered list with 'Research' as a single step within the 'Activity planning' cycle was unacceptable to about half of co-authors.
- Having two lists was strongly rejected by some as it did not highlight the inter-connected nature of 'Research' and 'Activity'.
- Necessary elements to communicate were felt to be
    - A clear recognition (without judgement of this) that authors likely come from a background in with 'Research' or 'Activity' are considered primary, or at least are the default conceptual frame of the author.
    - Inter-connectedness between 'Activity Planning' and 'Research', with (i) overlap (ii) two-way exchange.
    - Cyclicity, particularly in 'Activity Planning' (i.e. reflecting and reviewing leads to new or inspires new activities - explicitly highlighted in point vii). Research also often inspires future research. Ideally, the research process would loop around and feed back into the activity planning, so we feel a cyclic arrow is also justified here.
    - Parity between 'Activity' and 'Research' (i.e. neither should necessarily be seen as a sub-activity of the other, and an approach by authors with an emphasis on either should be seen as valid).

Technical corrections

Line 56: "video games" (a space is missing)
> Changed. Although online and magazine usage has changed, dictionaries keep two words.

Line 277: "is a risk"
> Changed.

Line 355: "a simple survey" missing indefinite article
> Changed.

Line 359: "a direct feed" missing indefinite article
> Changed.

Line 419: "school children" space missing
> In various dictionaries, 'schoolchildren' as a single word seems to be preferred.

Line 437: "make" remove the s
> Changed.

Line 510 "recommended" add ed
> Changed.

**RC2 - Louise Arnal**

This GC editorial builds on the first GC editorial by Illingworth et al. (2018), and provides a detailed route to publication aimed at geoscientists involved in geoscience communication activities. I found this editorial very insightful and a good balance between theory and illustrative examples of impactful GC publications. I wish I had read this editorial at the start of my SciComm/SciArt career during my PhD!
> Thank you.

Please find a few minor points below which will hopefully help improve this editorial for publication.

RC2.1 - P3 L74-77: The phrasing of these sentences makes the two first items: "complying with funders' requirements" and "communicating with relevant stakeholders", almost secondary and readers might dismiss them. I would suggest rephrasing the sentences to highlight the importance of all of these three valid points, and explicitly linking to sections of the paper that describe these points in more details.
> Point taken. The phraseology may not be as clear as it could be, and this could move beyond emphasis to make interpretation difficult for non-native speakers.  Rephrased.

RC2.2 - P3 L76-77: I found reading this sentence about contributing to building a field of geoscience communication a little bit intimidating. The first thought I had was that as I didn't get any training as a geoscience communicator, am I still entitled to contributing to the field's literature? You tackle this point really well later in the paper when you talk about collaborating on geoscience communication activities and outputs, but I was wondering if it might be helpful to allude to this already now, for readers like me?
> Altered text, adding " by which both new and experienced communicators contribute " to make it clear that all are entitled to contribute.

RC2.3 - Section 3: Another approach I have seen many geoscientists follow is a mix of both approaches illustrated in Fig. 1, where the activity design is done following approach 1a, and later reframed to publish it following approach 1b. I was wondering if you could comment on this and whether it is desirable?
> The main text and figure caption has been modified to explicitly state that these are end-member approaches.  This allows for your observation of a mixture of the two approaches. In any research *post-hoc* re-casting for publication is not ideal, although we suspect from our own experiences it is quite common.

RC2.4 - P5 L133-135: I suggest changing "useful" to "applicable" or "impactful". The outcomes might probably still be useful for a certain end, but it might be harder to draw any impact retrospectively.
> Thank you for this comment.  We believe that 'potential less useful' allows for outcomes that might still be useful for some purposes.

RC2.5  - P5 L152-154: Investigations of the dialogue and communication process is a great idea! Could you given an example or two of papers that do this well? A minor additional comment, I found this point slightly out of place here. Consider moving it (along with other recommendations) to a "further recommendations" section at the end of the paper if you think it would work well.
> As pointed out by Reviewer 1, our phraseology inadvertently implied that there were no examples of this in *GC*. That would have been out of place. We have changed the phraseology to clarify that there are some examples of this in GC, but we might like to see them more frequently. This both provides examples, and justifies the current location of the lines.

RC2.6 - P7 L199-201: I found defining what success looks like a very hard task when writing my first SciArt project proposals, and very different to anything I had written in science before (especially as an early career scientist who had never written a grant application). In science, "success" is very abstract in that an experiment might be successful either if it fails or if it works as expected, because both outcomes are scientific findings in their own right. Could you maybe elaborate a bit more on this task and/or provide some useful literature/resources on the topic to guide readers?
> Thank you. We have elaborated on this a little more, providing a reference to literature (a book) to provide further guidance for readers.

*Measuring success is largely based on two questions that you need to address at the start of any initiative: 'what' are your aims and objectives, and 'who' is your audience (Illingworth and Allen, 2020). In answering these question an aim can is 'what you want to achieve', while an objective should be thought of as 'the action(s) that you will take in order to realise an aim'. Each objective should be tied to a specific aim, and should also be SMART, i.e., Specific, Measurable, Achievable, Realistic, and Time-bound. Reflecting on the extent to which you have achieved these aims and objectives (ideally by using a reflective model; see Section 8.2) will help you to measure your success and also to better understand why certain aims and objectives were not met and what the result of this was for the initiative.*

RC2.7 - P7 L212: Maybe reiterate here that they can be quantitative or qualitative data and give a few quick examples?

> Thank you for this suggestion. Reiterating certain aspects such as data being quantitative or qualitative, or giving examples would raise questions as to why these and not more or all aspects of data had been highlighted again. And, examples are given in Section 6 immediately below. So, we have not expanded 'Collect data'.

RC2.8 - P14 L428: Could you please define here what "network analysis" means?

> To remain concise, we refrain from adding descriptions of the methods noted.  Instead, we provide a citation so that the reader can investigate further if they wish.

RC2.9 - P15 L469: I would add that these forms can usually be found directly via one's institute, for readers wondering where to find them.

> Text modified.

RC2.10 - Section 10: Here, you focus on GC publication as it is the target of this editorial. It might be worth noting here that widely accessible communication of research can also be achieved in different spaces using various formats to reach specific audiences, and that publishing in GC is the space and format you focus on here. E.g., Exhibition visitors who might not necessarily know about GC might find it interesting to find out about geoscientists' analysis of an exhibition via blog posts, a series of posts on social media, short videos, etc.

> Thank you. We have added a sentence to reflect this further, onward communication.

RC2.11 - Section 11: Are the points in this section in a specific order? I would swap some of them around (e.g., 2 and 3 before 1), so this led me to wonder if these were in a particular order.

> They are in partial order. Point 1 is first as this is our key desire for the reader to understand i.e. please plan (before doing something) if you want a smooth route to publication. Point 7 is at the end to fit with the paper's structure.  The rest a of somewhat equal importance, and can be seen as more or less important depending on the background of the geoscience communicator (specifically the co-authors).

RC2.12 - Figure 3: Consider adding a legend of what the different dots are on this figure. What is the fourth dot?

> We have changed the figure to remove the dots to avoid confusion. They were illustrative only because, as noted in the text, the positioning of the dot for any project is a subjective judgement of the investigators.

*Technical corrections:
- additional comma not needed before parenthesis is being closed. - brackets not needed around "Hut et al.".
> Changed.

- "to" can be removed.
> Please clarify which line this is on.

- "ed" missing at the end of "recommend".
> Changed.

**CC1 - Rhian Salmon**

CC1.1 - This paper provides a useful encouragement for any prospective contributors to Geoscience Communication. It is primarily focused on the criteria and approaches that are likely to lead to successful publication in this journal. It does, however, seem to gloss-over what many would argue to be the hardest part of this kind of work, namely, analysing the data.
> Thank you. We respond to your comment about analysing the data in CC1.3 below.

CC1.2 - On page 7, a simple 6-point process is described. While I agree that it's critical to define "what success looks like", I would argue that analysis against this criteria alone will lead to an "evaluation" rather than a research paper. A research paper, more often than not, will have a deeper question beyond simple evaluation against a pre-defined success metric. Some explanation about the difference between these would be helpful.
> We agree that 'what success looks like' is related to evaluation, and realise that the previous phraseology was overly focussed on 'evaluation'.  This we did not consciously intend, and used your words about a deeper question to modify point 1 of the list in order to clarify that is only one sort of research question out of a rich landscape of potential questions.

CC1.3 - It was also surprising to me that no further padding was included around Step 5 "Analyse the data" (line 213). This is surely the hardest area for someone who has not been trained in these methodologies, and the part of the process where guidance and collaboration might be most helpful. The subsequent case studies provide excellent examples related to the level of specific expertise that might be required for this step, and section 8.2 expands on this a bit more, but it might be worth adding at least a sentence at this early stage indicating that this step requires particular research expertise and a substantial amount of work!
> Two sentences added to emphasise these points.

CC1.4 - Figure 2 provides an interesting approach to conceptualising the research planning framework, which I found helpful while reading the text. Two design suggestions related to this figure:
- the grey box in the middle I think needs to be labelled (v) rather than (iv) with reference to the stages on the left hand side (purple);
> Thank you. Yes. This has been changed.
- I think it would be more compelling if the grey box ALSO correlated with the cycle on the right hand side (green). This could be achieved if the green cycle was a mirror-image to the purple one, ie, running anticlockwise, with both cycles overlapping and passing through the box in the middle named "plan and undertake research-informed communication". Currently, it looks like that happens either before the research question is defined, or after the paper has been written.
> Thank you. We agree that it would be good to make it appear that the grey box was integral to, or expands to become, the cycle on the right.  We have explored ways of doing this, including your suggestion. We have also tested different colours and found that displaying the figures in greyscale works very well. This will also aid readers with colour-blindness, and those who wish to print in black & white.  The proposed figure is now simplified, and includes a bold black outline to relate box (v) more obviously to the research process on the right without making it appear to be an element within in.

CC1.5 - I was also surprised by the narrative related to how high or low stakes a particular initiative might carry. This appears initially at line 234, later at 349, and then is expanded in figure 3. While I appreciate that science communication research might require different amounts of rigour and depth depending on the outcome and impact, I think it is risky to infer that it's ok if some ("low stakes") science communication research might not need specific skillsets for their data analysis, and therefore might not "warrant wider interdisciplinary input".
> We quite firmly and explicitly suggest collaboration (e.g. L375 of reviewed text).  The point here is that mistakes will have fewer implications so may be a good entry point for new workers learning skills.  We feel that it is important that geoscientists do not need to necessarily feel they have to build a large team to do geoscience communication - or this starts to form a substantial barrier.

CC1.6 - I'm not entirely convinced of the value of Figure 3 overall. In addition, the relevance of the placement of the various dots is not clear from either the caption or the text – if they refer to specific case studies discussed in the paper then they need to be appropriately labelled or identified.
> To avoid confusion, we have amended the figure to remove the dots.

CC1.7 - Finally, the paper provides a useful overview of the methods that have been used to date in GC articles. I wonder, however, if the purpose of this article is to encourage greater breadth in submissions. If so, it might be worth noting that there are several additional approaches, and types of data, that may be used for documenting and publishing communication work, such as think-pieces, auto-ethnographic works or explorations using art and other creative processes. It's not clear from this paper if such articles would indeed be welcomed by GC – they certainly would present a different kind of "data" as that explored here. I would recommend making this clear either way (and , if not welcome, suggesting that such papers would be better suited to alternative journals focused on public engagement with science).
> Thank you. This paper outlines how prospective authors might consider turning their science communication and public engagement activities into publishable research, presenting several examples of research methods that have done so successfully in *Geoscience Communication*. However, we acknowledge that there are many other research methods that researchers might wish to utilise, including (but not limited to) autoethnographies, walking interviews, and discourse analysis. Many of these research methods have a trusted provenance in other disciplines such as social sciences and pedagogy but might be alien to researchers who have initially trained in the geosciences. For those researchers who are keen to try out some of these methods for themselves, we encourage them to both collaborate with experts from these other disciplines and also to make use of the new *GC Insight* manuscript type, which has been specifically designed to present innovative and well-founded ideas related to geoscience communication, which have not yet been comprehensively explored, in a concise way.

CC1.8 - Finally, it would be useful to also include a short section outlining the level of support that GC provides during the submission and review process. For example, is there a pre- submission "pitch" stage, do you offer suggestions for potential collaborators, what is the peer-review process, and what is your recommended approach to co-authorship. This things may differ from the main discipline in which the prospective authors or communicators are familiar with.
> The support provided by GC is described in the first editorial https://gc.copernicus.org/articles/1/1/2018/ . It is *ad hoc*, based on conversations with the editorial team, rather than through a formalised process. It is therefore not something we can elaborate further on at this time. We will consider this further as an editorial team.

Despite these comments, I think it's great that this paper has been drafted and hope it will encourage further publication on this field.
> Thank you.

**Editorial: Geoscience communication - Planning to make it publishable**

[revised manuscript text omitted]

“An effect on, change or benefit to the economy, society, culture, public policy or services, health, the environment or quality of life, beyond academia”.
* * *
**Commented [JH5]:** Sentence modified in response to RC1.10

**Commented [JH6]:** RC2.1 - First sentence rephrased and simplified.

**Commented [JH7]:** RC1.11

**Commented [JH8]:** RC2.2

[revised manuscript text omitted]

**Commented [JH11]:** 3 sentences added in response to RC2.6

**Commented [JH12]:** CC1.2

**Commented [JH13]:** RC1.13

**Commented [JH14]:** RC1.14

[revised manuscript text omitted]

Commented [JH32]: Dots removed - CC1.6, RC1.12